# The *S.* Typhi effector StoD is an E3/E4 ubiquitin ligase which binds K48- and K63-linked diubiquitin

Melanie A McDowell[1],*, Alexander MP Byrne[2],*, Elli Mylona[2], Rebecca Johnson[2], Agnes Sagfors[2], Valerie F Crepin[2], Susan Lea[1], Gad Frankel[2]

*Salmonella enterica* (e.g., serovars Typhi and Typhimurium) relies on translocation of effectors via type III secretion systems (T3SS). Specialization of typhoidal serovars is thought to be mediated via pseudogenesis. Here, we show that the *Salmonella* Typhi STY1076/t1865 protein, named StoD, a homologue of the enteropathogenic *Escherichia coli*/enterohemorrhagic *E. coli*/*Citrobacter rodentium* NleG, is a T3SS effector. The StoD C terminus (StoD-C) is a U-box E3 ubiquitin ligase, capable of autoubiquitination in the presence of multiple E2s. The crystal structure of the StoD N terminus (StoD-N) at 2.5 Å resolution revealed a ubiquitin-like fold. In HeLa cells expressing StoD, ubiquitin is redistributed into puncta that colocalize with StoD. Binding assays showed that StoD-N and StoD-C bind the same exposed surface of the β-sheet of ubiquitin, suggesting that StoD could simultaneously interact with two ubiquitin molecules. Consistently, StoD interacted with both K63- ($K_D$ = 5.6 ± 1 $\mu$M) and K48-linked diubiquitin ($K_D$ = 15 ± 4 $\mu$M). Accordingly, we report the first *S.* Typhi–specific T3SS effector. We suggest that StoD recognizes and ubiquitinates pre-ubiquitinated targets, thus subverting intracellular signaling by functioning as an E4 enzyme.

## Introduction

*Salmonella enterica* subspecies *enterica* is divided into typhoidal (e.g., *S.* Typhi and *S.* Paratyphi) and non-typhoidal serovars (e.g., *S.* Typhimurium and *S.* Enteritidis). *S.* Typhi, the causative agent of typhoid fever, is a human-restricted pathogen, which is estimated to cause more than 20 million cases per year, resulting in 100,000–200,000 deaths (1, 2).

Central to *S. enterica* virulence is the function of two type III secretion systems (T3SS) encoded on *Salmonella* pathogenicity islands 1 and 2 (SPI-1 and SPI-2), which secrete effectors that subvert host cell processes during infection (3). The SPI-1 T3SS is active when *Salmonella* are extracellular, where it functions to allow invasion of non-phagocytic host cells, whereas the SPI-2 T3SS is activated upon internalization, where it functions to maintain a stable and permissive intracellular niche termed the *Salmonella*-containing vacuole (3). In *S.* Typhimurium, more than 40 effectors have been described, but this effector repertoire is reduced in *S.* Typhi, where approximately half are either absent or pseudogenes: SopA, SopE2, GogA, GogB, SopD2, SseI, SseJ, SseK1, SseK2, SseK3, SpvB, GtgA, CigR, SrfJ, SlrP, AvrA, SspH1, SteB, SteE, and GtgE, as well as the plasmid-encoded effectors SpvB and SpvC (4, 5). Other effectors appear to be "differentially evolved" between the typhoidal and non-typhoidal serovars, including SipD, SseC, SseD, SseF, SifA, and SptP (6, 7).

Although for many years, *S.* Typhi pathogenesis has been modelled using *S.* Typhimurium, it is now apparent that these serovars use distinct infection strategies. We have recently reported that whilst exposure to 3% bile triggers expression of SPI-1 genes and invasion of non-phagocytic cells in *S.* Typhi, it had an opposite effect in *S.* Typhimurium, resulting in repression of SPI-1 gene expression and invasion (8). Moreover, expression of the *S.* Typhimurium T3SS effector GtgE in *S.* Typhi allows it to replicate within nonpermissive bone marrow-derived murine macrophages because of the proteolytic activity of GtgE on Rab32 (9). In contrast, *S.* Typhi encodes the virulence factors Vi-antigen and typhoid toxin, which are absent from *S.* Typhimurium (4, 10, 11), suggesting that *S.* Typhi may encode other, serovar-specific virulence factors yet to be identified.

Recently, while searching for paralogues of the enteropathogenic *Escherichia coli* (EPEC) T3SS effector NleG, we identified an open reading frame, *STY1076* (*S.* Typhi CT18)/*t1865* (*S.* Typhi Ty2) that is absent from *S.* Typhimurium. NleG effectors share a conserved C-terminal U-box E3 ubiquitin ligase domain that engages with host ubiquitination machinery and have highly variable N-terminal regions presumed to be involved in substrate recognition (12). Recently, the MED15 subunit of the Mediator complex has been identified as a target of the enterohemorrhagic *E. coli* (EHEC) effector NleG5-1, whereas hexokinase-2 and SNAP29 are targeted by

---

[1]Sir William Dunn School of Pathology, University of Oxford, Oxford, UK   [2]MRC Centre for Molecular Bacteriology and Infection, Department of Life Sciences, Imperial College, London, UK

Correspondence: g.frankel@imperial.ac.uk; susan.lea@path.ox.ac.uk
Melanie A McDowell's present address is Biochemistry Centre (BZH), University of Heidelberg, Heidelberg, Germany
Alexander MP Byrne's present address is Avian Virology and Mammalian Influenza Research, Virology Department, Animal and Plant Health Agency, Surrey, UK
*Melanie A McDowell and Alexander MP Byrne contributed equally to this work

 

NleG2-3 (13). The aim of this study was to determine whether *STY1076* is a T3SS effector and to elucidate its structure and function.

# Results

## The *S.* Typhi outer protein D (StoD)

Since first identified as T3SS effectors in the mouse pathogen *Citrobacter rodentium* (14), NleG proteins have been found in EPEC and EHEC (15), as well as *Salmonella bongori*, where it is named SboD (*S. bongori* also contains two truncated NleG family members named SboE and SboF) (16). Interestingly, a homologue of SboD is found in *S.* Typhi (*STY1076* in the CT18 strain; *t1865* in the Ty2 strain), but not *S.* Typhimurium or *S.* Enteritidis (16). We renamed *STY1076/t1865*, which is located at the distal part of phage ST10 of *S.* Typhi CT18 (Fig 1A), StoD, in keeping with the *S. bongori* nomenclature. A StoD homologue is also present in *S.* Paratyphi B, *SPAB_02256*, here renamed as *S.* Paratyphi B outer protein D (SpoD), in keeping with this nomenclature.

The overall sequence identity of StoD compared with other NleG proteins ranges from 25.4% (EPEC NleG) to 74.66% (*S. bongori* SboD). Sequence alignment revealed that the N-terminal region shows varying homology, ranging from 9.52% (*C. rodentium* NleG1) to 69.17% (*S. bongori* SboD) (Fig S1). In contrast, the C termini are more homologous to each other with sequence identity ranging from 37.62% (EHEC NleG 2-2 and *C. rodentium* NleG8) to 82.18% (*S. bongori* SboD) compared with StoD. The C terminus of StoD contains conserved residues for a U-box–type E3 ubiquitin ligase domain, in particular three residues shown to be involved in binding to E2 ubiquitin–conjugating enzymes: V165, L167, and P204 (12) (Fig S1). The evolutionary history of the NleG proteins (Fig 1B) shows that the *Salmonella* NleG–like effectors cluster into a separate clade. This suggests that the *Salmonella* proteins evolved from an ancestral protein shared with some of the *E. coli* and *C. rodentium* effectors, before diverging into the different *Salmonella* species and serovars.

## StoD is a SPI-1 effector

Considering that *S. bongori* only expresses SPI-1, we aimed to determine if StoD is an SPI-1 *S.* Typhi effector. Because of safety constraints of working with *S.* Typhi, secretion and translocation assays were carried out using *S.* Typhimurium as a surrogate. To this end, we transformed WT *S.* Typhimurium and a Δ*prgH* mutant with a plasmid encoding StoD from its endogenous promoter with a 4xHA C-terminal tag. Endogenous SipD, an SPI-1 T3SS translocator (17), was used as a positive control, whereas the cytosolic protein DnaK was used as a lysis control. Western blotting of bacterial pellets (protein expression) and culture supernatant (protein secretion) revealed strong expression of StoD, SipD, and DnaK in the pellets. SipD and StoD were detected in the supernatants of WT *S.* Typhimurium but not of Δ*prgH* (Fig 1C), suggesting that StoD is secreted via the SPI-1 T3SS.

We next used the β-lactamase translocation assay (18) to assess if StoD is translocated into host cells. *stoD* and the SPI-1 control

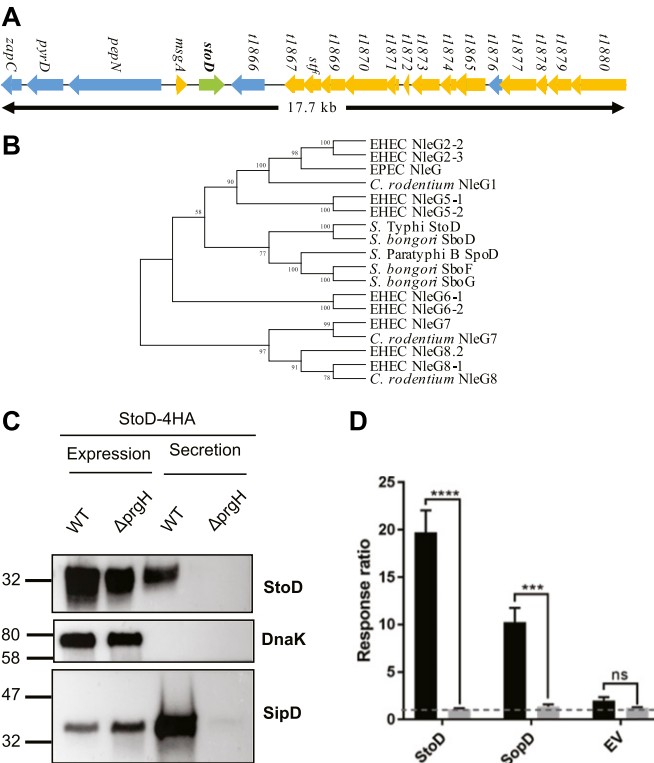

**Figure 1. StoD is a member of the NleG family of effector proteins.**
**(A)** A diagrammatic representation of the genomic localization of *stoD* within the *S.* Typhi Ty2 genome. Colours indicate different gene functions: phage genes (yellow), *stoD* (green), and miscellaneous genes (light blue). **(B)** The evolutionary history of the NleG family members from EHEC, EPEC, *C. rodentium*, *S. bongori*, *S.* Typhi, and *S.* Paratyphi B. **(C)** Secretion assay of 4HA-tagged StoD from WT and Δ*prgH S.* Typhimurium; SipD and empty pWSK29-Spec vector (EV) were used as positive and negative controls, respectively. DnaK was used as a lysis and loading control. An anti-HA antibody was used to detect HA-tagged StoD. SipD and DnaK were detected using respective antibodies. The blot is representative of two repeats. **(D)** HeLa cell translocation of StoD-TEM1 and SopD-TEM1 fusions from WT or Δ*prgH*Δ*ssaV S.* Typhimurium; empty pWSK29-Spec vector (EV) was used as a control. Graph shows mean + SEM. Translocation of each protein was compared between the WT and Δ*prgH*Δ*ssaV* genetic backgrounds using a Multiple *t* test with the Holm-Sidak correction for multiple comparisons (****$P < 0.0001$). Graph represents an average of three independent repeats.

effector *sopD* from *S.* Typhi Ty2 were cloned into the pWSK29-Spec vector (7) with a C-terminal β-lactamase (TEM1) fusion. The plasmids encoding the TEM1-tagged effectors were transformed into WT *S.* Typhimurium and a double Δ*prgH*Δ*ssaV* mutant, deficient in translocation via both *SPI-1* and *SPI-2* T3SSs (19, 20). At 3 h post-infection, both SopD-TEM1 and StoD-TEM1 were translocated by WT but not by Δ*prgH*Δ*ssaV S.* Typhimurium, indicating that they are translocated in a T3SS-dependent manner (Fig 1D).

Functional assays revealed that StoD plays no role in *S.* Typhi invasion into HeLa cells or replication in the macrophage-like THP1 cells (Fig S2).

## StoD is an E3 ubiquitin ligase

Because StoD was originally identified as a homologue of NleG and was predicted to have a U-box E3 ubiquitin ligase domain (Fig S1), we investigated if it has E3 ubiquitin ligase activity. We used

recombinant StoD in autoubiquitination assays, a method used to determine E3 ubiquitin ligase activity in the absence of a known substrate (21). Recombinant StoD was combined with an E1 ubiquitin-activating enzyme (UBE1), ubiquitin, ATP, and a range of E2 ubiquitin–conjugating enzymes (UBE2K, UBE2H, UBE2R1, UBE2D1, UBE2D2, UBE2D3, UBE2E1, UBE2L3, UBE2E3, UBE2C, and UBE2N); autoubiquitination was assessed by Western blotting. As StoD was most active in the presence of UBE2E1 (Fig S3), it was used in the functional and structural studies described below.

It has previously been shown that L123K substitution in NleG2-3 did not affect the autoubiquitination activity, whereas a P160K substitution inactivated the ligase (12). As StoD possesses equivalent leucine and proline residues at positions 167 and 204, respectively, we investigated the effect of L167A and P204K substitutions on the activity of StoD. Ubiquitination assays revealed that, similarly to NleG2-3, $StoD_{L167A}$ was biologically active (Fig 2A), whereas $StoD_{P204K}$ was inactive (Fig 2B).

We next investigated which domain of StoD was required for autoubiquitination. $StoD_{1-95}$ (StoD-N) and $StoD_{134-233}$ (StoD-C) were combined with UBE1, ubiquitin, ATP, and UBE2E1, and autoubiquitination

was assessed by Western blotting. Autoubiquitination was observed in the presence of StoD-C, but not StoD-N, confirming that only the C terminus of StoD has autoubiquitination activity (Fig 2B). Similar to full-length $StoD_{P204K}$, $StoD-C_{P204K}$ exhibited no autoubiquitination inactivity (Fig 2B).

To characterize the interaction between StoD and E2 ubiquitin–conjugating enzymes by NMR spectroscopy, we assigned all non-proline backbone amides of $StoD-N_{1-101}$ (Fig S4A) and StoD-C (Fig S4B) in their $^1H$, $^{15}N$-HSQC spectra and titrated the domains with UBE2E1. The StoD-N spectrum showed no chemical shift perturbations (CSPs) with a 3 molar excess of UBE2E1 present (Fig S5), whereas the StoD-C spectrum exhibited clear CSPs and line broadening in an equimolar titration (Fig S6A), confirming that the interaction with E2 ubiquitin–conjugating enzymes is confined to StoD-C. The observed CSPs were then mapped on to the surface of a SCWRL homology model (22) for StoD-C, constructed from the solution structure of the NleG2-3 C terminus (38.61% sequence identity; Fig S1) (12). The CSPs mapped to the common E2-binding site within the core U-box motif (Fig 2C) (12), equivalent to that observed in the structure of the CHIP E3 ligase U-box in complex

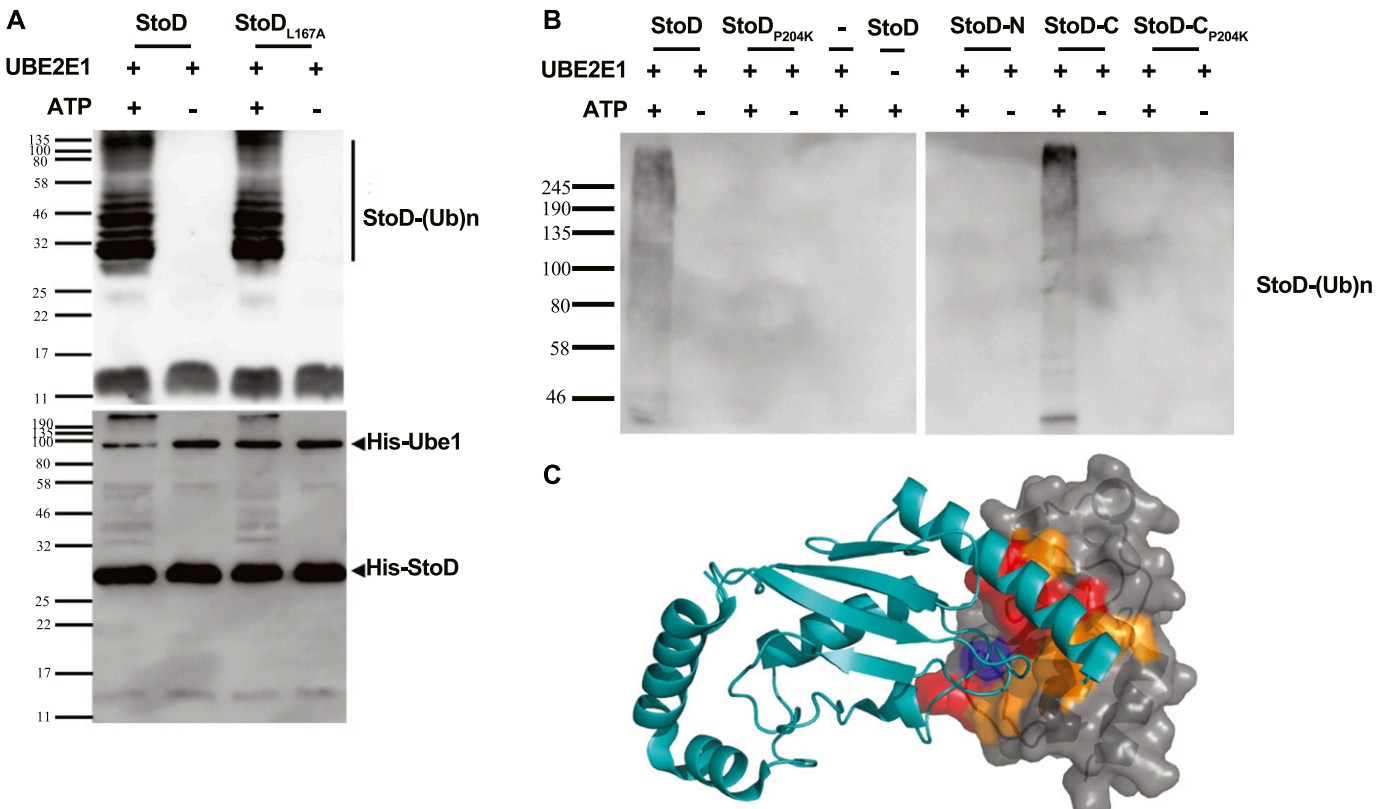

**Figure 2. StoD is an E3 ubiquitin ligase.**
**(A)** Both StoD and $StoD_{L167A}$ have an E3 ubiquitination activity in the presence of ATP (upper panel shows an anti-Ub-FK2 antibody blot). Western blotting using anti-His tag antibodies shows autoubiquitination of StoD (lower panel). **(B)** Autoubiquitination assay using StoD, $StoD_{P204K}$, StoD-N, StoD-C, or $StoD-C_{P204K}$ visualised with anti-Ub-FK2 antibody. Only StoD and StoD-C exhibit an E3 ubiquitin ligase activity. Image is representative of two independent repeats. **(C)** Model for the interaction between StoD-C (grey surface) and UBE2E1 (cyan cartoon) based on the CHIP U-box/UBE2D2 structure (23) (PDB ID 2OXQ) constructed using Superpose (68). A SCWRL homology model (22) for StoD, constructed using the sequence alignment in Fig S1 and solution structure of reduced NleG2.3 (12) (PDB ID 2KKX), was superimposed with CHIP U-box (RMSD 2.07 Å over 55 residues). The UBE2E1 structure (69) (PDB ID 3BZH) was superimposed with UBE2D2 (RMSD 0.61 Å over 149 residues). These superimpositions are shown in Fig S6C. CSPs from titration of 100 µM $^{15}N$-StoD-C [134–233] with 100 µM UBE2E1 are mapped onto the surface of StoD-C: peak disappearances due to line broadening are shown in red, whereas peak shifts greater than 0.05 ppm are shown in orange. P204 is shown in blue.

with UBE2D2 (Fig S6C) (23). Despite being absent from ¹H, ¹⁵N-HSQC spectra, P204 is also likely to contribute to this interaction interface (Fig 2C), as suggested by the autoubiquitination assay (Fig 2B). Furthermore, alanine mutation of L167, also found within this binding surface (Fig S6D), leads to only minor changes in the structure of StoD-C and yet was already sufficient to abolish the interaction with UBE2E1 (Fig S6B). As StoD$_{L167A}$ is still capable of autoubiquitination when present in excess of the E2 enzyme (Fig 2A), this mutant may still undergo a weak interaction with UBE2E1 that is not observable in the equimolar NMR titration. Taken together, these results indicate that StoD-C represents a canonical U-box E3 ligase domain.

### The structure of StoD-N reveals a ubiquitin-like fold

Although StoD-N is thought to be important for substrate recognition (12), the divergent sequence of this domain (Fig S1) makes it difficult to predict structure and function. Therefore, residues 1–101 of the domain were expressed with an N-terminal tag in *E. coli*, purified to homogeneity and crystallised as native and seleno-methionine derivatives. The 2.5 Å resolution crystal structure of StoD-N (Figs 3A and S7) was subsequently determined by anomalous dispersion (Table 1). The last six amino acids of the domain are disordered and not included in the structure. StoD-N crystallised as a tetramer; however, inter-subunit contacts are predominantly formed by the N-terminal tag (Fig S8), suggesting this is not a physiologically relevant oligomer. Indeed, size exclusion chromatography with in-line multiangle light scattering (SEC-MALS) of untagged StoD, StoD-N, and StoD-C showed that both the full-length protein and individual domains are monomeric in solution up to 16 mg/ml (Fig S9). Despite two loop regions having higher B-factors (Fig S10A), the four subunits of the crystallographic tetramer superimpose with an average root mean square deviation (RMSD) of 0.34 Å (Fig S10B), indicating StoD-N has a relatively rigid structure.

StoD-N has a globular α/β sandwich fold, comprising two α-helices packed against a twisted four-stranded β-sheet, with one parallel and two antiparallel β-strand interactions (Fig 3A). The structure is highly similar to the recent structure of the N-terminal domain of NleG5-1 (13) (Fig S11A), despite their low sequence similarity (21%). This suggests that StoD and the NleG family members likely have a conserved N-terminal structural fold, with variability in surface residues allowing ubiquitination of distinct targets. A search of the PDB with our structure coordinates using the DALI algorithm (24) revealed the complete StoD-N and NleG5-1 domains to have a novel fold, whereas the β-sheet and first α-helix show structural homology predominantly to two domains of known function. First, these secondary structure elements align with those of SH2 domains (Fig S11B), which act as phosphotyrosine (Tyr(P))-binding modules. However, the long-kinked C-terminal helix of StoD-N, a unique feature of this domain, occludes the common Tyr(P)-binding site (Fig S11B). Furthermore, the ¹H/¹⁵N-HSQC of StoD-N showed no CSPs when titrated with an 80 molar excess of Tyr(P) (Fig S11C), indicating StoD-N is unlikely to be involved in phospho-recognition. Second, ubiquitin has structural homology with StoD-N, which is again made structurally distinct by the additional C-terminal helix (Fig 3B). Therefore, StoD-N can be defined as a new ubiquitin-like (Ubl) domain, which is striking in the context of the role of the full-length protein as an E3 ligase, which uses ubiquitin as a substrate.

### StoD colocalizes with and binds to ubiquitin

To gain further insights into the cellular function of StoD, we aimed to localize it during infection of cultured cells. However, we were unable to detect 4HA-tagged StoD translocated from *S*. Typhi (data not shown). For this reason, we transiently transfected HA-tagged StoD into HeLa cells and used anti-HA antibodies for localization by immunofluorescence. Transfection of the control HA-mCherry resulted in a diffuse localization throughout transfected cells. In contrast, transfected StoD-HA formed discrete puncta throughout transfected cells (Figs 4A and S12). These puncta did not colocalize with the common eukaryotic vesicular proteins Rab11a, Vamp3, or LC3 (Fig S13). As StoD is an E3 ubiquitin ligase, we investigated if the puncta seen during transfection is reflected by redistribution of cellular ubiquitin. StoD-HA was transfected into HeLa cells, and the localization of cellular ubiquitin was determined by immunofluorescence. Upon transfection of HA-mCherry as a control, cellular ubiquitin was seen throughout the cell with no distinguishable localization. In contrast, transfection of StoD-HA caused redistribution of ubiquitin into puncta that colocalized with StoD-HA in 61% of transfected cells (Fig 4B and C). Transfection of StoD-N caused redistribution of ubiquitin in 5.5% of transfected cells, whereas transfection of StoD-C did not cause redistribution of ubiquitin (Figs 4C and S12). Redistribution of the ubiquitin-like proteins SUMO-1, SUMO-2/3, or NEDD8 was not observed upon transfection of StoD-HA (Fig S14). Therefore, StoD is able to cause the specific relocalization of cellular ubiquitin upon transfection, either because it can bind ubiquitin or is itself heavily ubiquitinated with both the StoD-N and StoD-C domains working together for efficient redistribution.

We used the yeast two hybrid (Y2H) assay to test the hypothesis that StoD is able to interact with ubiquitin. Full-length StoD and StoD derivatives were cloned into pGBKT7 to generate a fusion with the DNA-binding domain of the transcriptional activator Gal4,

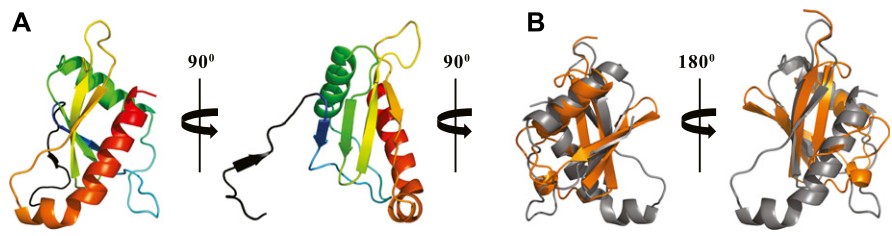

**Figure 3.  The StoD N terminus is a ubiquitin-like domain.**
**(A)** Views of the 2.5 Å crystal structure of the StoD-N [1–101] shown as a cartoon representation. Chain A is shown coloured from the N terminus (blue) to the C terminus (red). Residues visible from the N-terminal tag are coloured black. **(B)** Superimposition of StoD-N [1–101] chain A (grey) with human ubiquitin (orange) (28) (PDB ID 1UBQ) using Superpose (68). The RMSD is 3.40 Å over 38 residues.

**Table 1. Data collection, phasing, and refinement statistics.**

| | Native dataset | SeMet dataset |
|---|---|---|
| Data collection | | |
| Space group | P4$_3$22 | P4$_3$22 |
| Cell dimensions | | |
| a, b, c (Å) | 92.96, 92.96, 156 | 93.08, 93.08, 155.8 |
| α, β, γ (°) | 90, 90, 90 | 90, 90, 90 |
| Wavelength (Å) | 0.972 | 0.972 |
| Resolution (Å)[a] | 34.65–2.54 (2.61–2.54) | 32.48–2.86 (2.93–2.86) |
| No. unique reflections[a] | 23,299 (1,687) | 16,485 (1,172) |
| R$_{sym}$ or R$_{merge}$[a] | 0.097 (0.786) | 0.151 (0.869) |
| Average I/Iσ[a] | 30.8 (4.8) | 22.2 (4.5) |
| CC1/2[a] | 1.000 (0.958) | 0.999 (0.939) |
| Completeness (%)[a] | 99.8 (99.9) | 99.8 (99.7) |
| Redundancy[a] | 26.1 (27.4) | 25.7 (25.8) |
| Refinement | | |
| R$_{work}$/R$_{free}$ | 21.6/24.4 | |
| Ramachandran[b] | | |
| Allowed | 100% | |
| Favoured | 93.6% | |
| MolProbity Score[b] | 1.60 (99$^{th}$ percentile) | |
| No. atoms | | |
| Protein | 3,495 | |
| Water | 35 | |
| Average B factor (Å$^2$) | 54.8 | |
| R.M.S deviation | | |
| Bond length (Å) | 0.01 | |
| Bond angles (°) | 1.15 | |

[a]Values in brackets are for the highest resolution shell.
[b]Determined using MolProbity (60).

whereas ubiquitin was cloned into pGADT7 to generate a fusion with the activation domain of Gal4. Protein interactions were detected between ubiquitin and StoD, StoD$_{P204K}$, StoD-N, and StoD-C (i.e. growth and blue colonies in QDO). No interactions were seen between StoD-C$_{P204K}$ and ubiquitin or when StoD or ubiquitin were expressed in the presence of the control empty pGADT7 or pGBKT7 vectors (Fig 4D). This suggests that both the N- and C termini of StoD are ubiquitin-binding domains (UBDs), but the interaction of the C terminus may be dependent on either the interaction with the E2 or the correct fold of this E3 ubiquitin ligase domain.

We determined if the ability of StoD-N to bind ubiquitin is shared with other family members. To this end, we investigated whether full-length NleG7, NleG7$_{P177K}$, and NleG8 from *C. rodentium*, as well as their N termini (amino acids 1–97 and 1–109, respectively) bind ubiquitin using Y2H. This revealed that whilst full-length NleG7 and NleG8 bound ubiquitin, NleG7$_{P177K}$, NleG7-N, and NleG8-N did not bind ubiquitin (Fig S15). These results suggest that, whereas interaction with ubiquitin is conserved amongst NleG family members, the ability of the N terminus to bind ubiquitin is specific to StoD.

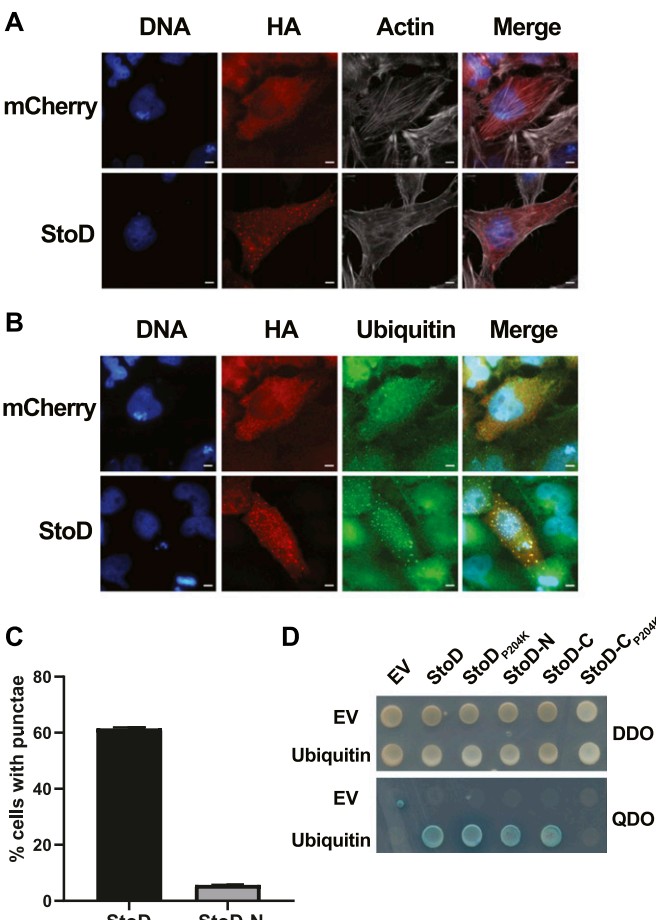

**Figure 4. StoD forms puncta upon ectopic expression which colocalize with cellular ubiquitin.**
**(A)** Immunofluorescence of HeLa cells transfected with HA-StoD reveals formation of discrete puncta. **(B, C)** Colocalization of transfected StoD and StoD-N with ubiquitin; no colocalization was seen in cell transfected with StoD-C or StoD$_{P204K}$ and the mCherry negative control. **(B)** DNA and actin were visualised using Hoechst 33258 and Phalloidin-iFluor 647, respectively. StoD-HA, StoD$_{P204K}$-HA, and HA-mCherry were visualised using an anti-HA antibody, whereas ubiquitin was visualised using an anti-Ub-FK2 antibody. Scale bar, 5 μm. Images representative of at least two independent repeats. **(C)** Percentage of transfected cells where colocalization of ubiquitin with either StoD or StoD-N is observed. **(D)** Direct Y2H assay in AH109 cotransformed with either empty pGBKT7 (EV) or ubiquitin and StoD derivatives. Cotransformants were plated on control DDO plates and QDO plates to assess protein–protein interactions. StoD, StoD$_{P204K}$, StoD-N (aa 1–133), and StoD-C (aa 134–223) interacted with ubiquitin. No interaction was seen in cotransformants expressing StoD-C$_{P204K}$ and ubiquitin. Image is representative of three independent repeats.

## StoD binds diubiquitin

We next investigated the interaction between StoD and ubiquitin in cell-free assays in vitro. Microscale thermophoresis (MST) was used to show that fluorescently labelled StoD and ubiquitin interact non-cooperatively with a K$_D$ of 43 ± 9 μM (Fig 5A). Fluorescence intensity measurements for the converse titration, using a G76C variant of ubiquitin to allow for C-terminal maleimide dye labelling, corroborated this interaction affinity (K$_D$ = 55 ± 11 μM) (Fig 5B). MST measurements with StoD-N and StoD-C individually confirmed that both domains interact directly with ubiquitin with a K$_D$ in the range

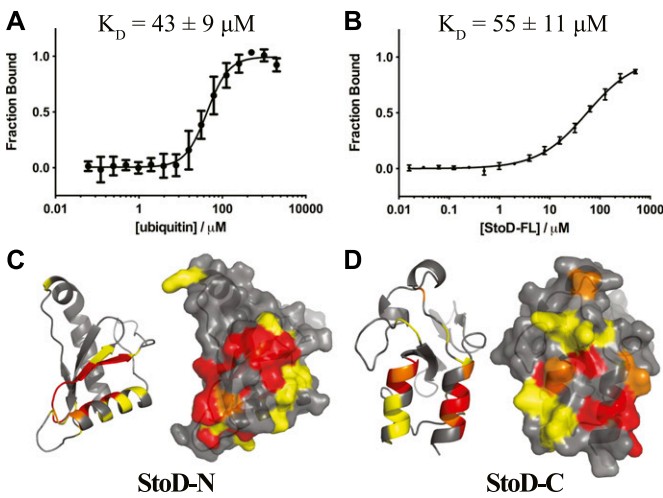

**Figure 5. StoD has two UBDs.**
**(A)** MST measured for a titration of 61 nM–2 mM ubiquitin with 40 nM fluorescently labelled StoD. 20% LED power and 40% laser power and data from the thermophoresis contribution alone were used. The normalized fluorescence signal was taken relative to that of the fully bound state and shown as an average of four independent dilution series. The data were fitted with a four parameter logistic (4PL) fit, yielding a Hill coefficient of 1.67 ± 0.23. **(B)** Fluorescence intensity measured for a titration of 16 nM–500 $\mu$M full-length StoD with 40 nM fluorescently labelled ubiquitin$_{G76C}$. The fluorescence signal was taken relative to that of the fully bound state and shown as an average of three independent dilution series. The data were fitted with a 4PL fit, yielding a Hill coefficient of 0.89 ± 0.06. **(C, D)** CSPs from titration of 100 $\mu$M $^{15}$N-StoD-N [1–101] or $^{15}$N-StoD-C [134–233] with 100 $\mu$M ubiquitin mapped onto the surface of the (C) StoD-N [1–101] crystal structure or (D) StoD-C [134–233] model shown in Fig 2C, respectively. Cartoon and surface representations of the same view are shown for clarity for each model. Peak disappearances due to line broadening are shown in red, peak shifts greater than 0.1 ppm are shown in orange, and those between 0.05 and 0.1 ppm are shown in yellow.

of 100 $\mu$M, despite the binding curves not reaching saturation (Fig S16). Therefore, StoD has a higher ubiquitin binding affinity than its composite domains, which is presumably avidity-mediated and likely explains why only the full-length protein efficiently re-distributes cellular ubiquitin (Figs 4C and S12).

To ascertain the molecular details of the StoD/ubiquitin interaction, $^{15}$N-labelled StoD-N and StoD-C were titrated with equimolar ubiquitin (Fig S17) and the resulting CSPs mapped onto our structural models for these domains (Fig 5C and D). In both cases, these localize to a defined surface, which for StoD-N comprises the N-terminal parallel $\beta$-strands and first $\alpha$-helix within the ubiquitin-like fold (Fig 5C). Notably, mutations in NleG2-3 that disrupt its interaction with hexokinase-2 also map to this $\beta$-sheet (13), indicating this may represent a common interaction surface for host target proteins. For StoD-C, the CSPs are confined to two $\alpha$-helices (Fig 5D), a surface that is notably distinct from the interface with UBE2E1 (Fig S18). Indeed, titration of $^{15}$N-labelled StoD-C with UBE2E1 and ubiquitin together showed characteristic CSPs for both binding partners and significant line broadening (Fig S18C), indicating the ternary complex had been formed in the solution. As $^{15}$N-labelled ubiquitin (Fig S19A) does not interact directly with an equimolar amount of UBE2E1 (Fig S19B), StoD-C is likely to be binding directly to both ubiquitin and UBE2E1 within this ternary complex. Therefore, the impaired interaction of StoD-C$_{P204K}$ with ubiquitin in Y2H is likely due to misfolding of this mutant.

Interestingly, this ubiquitin interaction site in StoD-C seems to be remote from the position of ubiquitin present in E2–Ub/RING E3 complex structures (25, 26) and actually faces away from the catalytic cysteine of UBE2E1 (Fig S18A). Furthermore, although ubiquitin is highly dynamic within the E2–Ub conjugate (27), it is unlikely the thioester-linked ubiquitin could adopt a position to reach this face of StoD-C without encountering steric hindrance (Fig S18B). Therefore, the data suggest that a separate ubiquitin moiety is bound by the identified surface of StoD-C, rather than ubiquitin in the context of the E2–Ub conjugate.

In a reciprocal experiment, $^{15}$N-labelled ubiquitin was titrated with an equimolar amount of either StoD-N, StoD-C, or full-length StoD (Fig S20), allowing CSPs to be mapped onto the surface of ubiquitin (28). Interestingly, both StoD-N and StoD-C interacted with the exposed surface of the $\beta$-sheet of ubiquitin (Fig 6A), which represents a known binding site for UBDs (29). Indeed, the hydrophobic residues L8, I44, and V70, which serve as the common binding platform, all undergo CSPs in the presence of either StoD domain.

As the linker connecting StoD-N and StoD-C is predicted to be disordered by the RONN algorithm (Fig S21A) (30), the two domains are likely capable of forming independent interactions with ubiquitin. Indeed, the $^1$H, $^{15}$N-HSQC spectrum of StoD overlays well

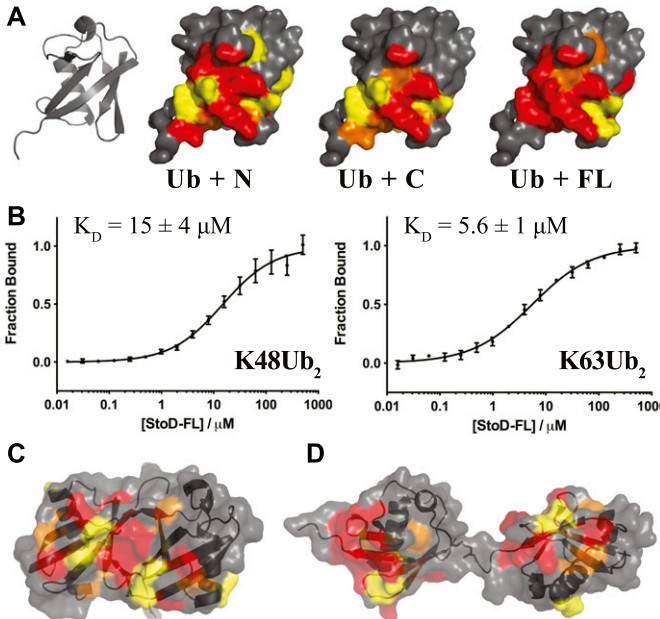

**Figure 6. StoD preferentially binds to diubiquitin.**
**(A)** CSPs from titration of 100 $\mu$M $^{15}$N-ubiquitin with StoD-N [1–101], StoD-C [134–233], or StoD-FL [1–233] mapped onto the surface of human ubiquitin (28) (PDB ID 1UBQ). Cartoon and surface representations of the same view are shown for clarity for each model. Peak disappearances due to line broadening are shown in red, peak shifts greater than 0.1 ppm are shown in orange, and those between 0.05 and 0.1 ppm are shown in yellow. **(B)** Fluorescence intensity measured for a titration of 16 nM–500 $\mu$M StoD with 40 nM fluorescently labelled K48-linked or K63-linked diubiquitin. The fluorescence signal was taken relative to that of the fully bound state and shown as an average of three independent dilution series. The data were fitted with a 4PL fit, yielding Hill coefficients of 0.88 ± 0.09 (K48Ub$_2$) and 0.77 ± 0.05 (K63Ub$_2$). **(C, D)** CSPs shown in (A) for StoD are mapped onto the surface of (C) K48-linked (33) (PDB ID 1AAR) and (D) K63-linked (34) (PDB ID 2JF5) diubiquitin.

with the individual spectra for StoD-N and StoD-C (Fig S21B), indicating the two domains do not interact and thus are unlikely to occlude each other's ubiquitin-binding sites (Fig 5C and D). Furthermore, as StoD-N and StoD-C bind to the same site on ubiquitin (Fig 6A), full-length StoD may be capable of interacting simultaneously with two molecules of ubiquitin. Therefore, we investigated if StoD had the ability to bind diubiquitin moieties with common isopeptide bond linkages. To this end, we performed controlled synthesis of K48-linked and K63-linked diubiquitin by combining distally blocked ubiquitin$_{K48R}$ or ubiquitin$_{K63R}$, respectively, with proximally blocked ubiquitin$_{G76C}$ (Fig S22), with the C-terminal cysteine enabling subsequent fluorescent labelling of one subunit with a maleimide dye. Fluorescence intensity measurements from titrations with StoD showed that the interaction affinity with K63-linked diubiquitin ($K_D$ = 5.6 ± 1 $\mu$M) was higher than with K48-linked diubiquitin ($K_D$ = 15 ± 4 $\mu$M) (Fig 6B). These interaction affinities are in line with those observed for similar ubiquitin-binding proteins; for example, the human proteasome receptor S5a, which also comprises two UBDs connected by a flexible linker, binds K48-linked diubiquitin ($K_D$ = 8.9 ± 0.6 $\mu$M) with a much higher affinity than monoubiquitin ($K_D$ = 73 $\mu$M) (31, 32). The ubiquitin-binding site for full-length StoD (Fig 6A) was then mapped onto the available structures of K48-linked (33) and K63-linked diubiquitin (34). Interestingly, this showed that in K48-linked diubiquitin, the two ubiquitin molecules assume a "closed" conformation, where the StoD-binding regions of both molecules are occluded between the two ubiquitin molecules (Fig 6C). In contrast, K63-linked diubiquitin assumes an "open" conformation, which exposes the StoD-binding regions of both ubiquitin molecules (Fig 6D) and may explain why StoD binds to this diubiquitin variant with a higher affinity (Fig 6B). Taken together, these results indicate that StoD has two UBDs that preferentially bind diubiquitin over monoubiquitin, with K63-linked diubiquitin being engaged three times stronger than K48-linked diubiquitin.

## Discussion

Although the closely related pathogens *S.* Typhimurium and *S.* Typhi both use two T3SSs to translocate effector proteins into eukaryotic cells, the host range and disease outcome are remarkably distinct. Despite this, much of the work in identifying and characterizing the *Salmonella* T3SS effector repertoire has been performed in *S.* Typhimurium and simply extended to *S.* Typhi, where the function of these effectors has been assumed to be the same. However, many of these effectors are either pseudogenes or completely absent from the *S.* Typhi genome (4) and until now, no attempt has been made to identify effectors that are unique to *S.* Typhi. In this study, it was found that *S.* Typhi StoD, which is absent in *S.* Typhimurium, was translocated and secreted by the SPI-1 T3SS of *Salmonella.* In line with its previous identification as a putative member of the NleG family of T3SS effector proteins, we confirmed StoD is capable of performing autoubiquitination with several eukaryotic E2 ubiquitin ligase enzymes, similar to other NleG proteins (12). Whereas we found that the StoD-C domain has the key features of a U-box E3 ligase domain,

the crystal structure of the StoD-N domain revealed a Ubl fold that is conserved with NleG5-1 (13).

*Salmonella* encodes several T3SS effectors that are E3 ubiquitin ligases; however, none of these are members of the NleG family and several, SopA, SlrP, and SspH1, are absent or pseudogenes in *S.* Typhi (35, 36, 37). Therefore, StoD is the first NleG protein family member to be identified in *S. enterica* and is only present in two typhoidal serovars, *S.* Typhi and *S.* Paratyphi B. Because of the sequence diversity of the N-terminal domain of the NleG proteins, it has been suggested that this is involved in substrate recognition (12) and would, therefore, direct different NleG proteins to different host targets. Indeed, the N-terminal domains of NleG5-1 and NleG2-3 were recently found to selectively target MED15 and hexokinase-2, respectively, despite likely having a conserved structural fold (13). This is also seen with the IpaH family of T3SS E3 ubiquitin ligase effector proteins found in *Salmonella flexneri*, *Salmonella* (SspH1, SspH2, and SlrP), and *Pseudomonas aeruginosa* (38). All IpaH proteins share the same overall topology: an N-terminal leucine-rich repeat (LRR) domain and a C-terminal NEL E3 ubiquitin ligase domain; differences in the LRR domain determine substrate specificity (39) and enable different IpaH proteins to ubiquitinate different host proteins. For example, IpaH 4.5, IpaH 9.8, and IpaH 0722 all inhibit the NF-$\kappa$B pathway but achieve this by ubiquitinating different substrates (39). This likely explains why EHEC, *C. rodentium*, and *S. bongori* have multiple NleG proteins.

Interestingly, upon transfection into mammalian cells, we found that full-length StoD caused the specific redistribution of cellular ubiquitin, colocalizing into discrete puncta. Furthermore, StoD can directly bind to ubiquitin through both the N- and C-terminal domains, which only together lead to ubiquitin redistribution in vivo. Crucially, this binding surface on StoD-C is distinct from that of the E2 ubiquitin–conjugating enzyme. In addition, both domains recognize the same ubiquitin surface that is commonly used by other UBDs (29), suggesting that StoD binds to two separate molecules of ubiquitin. This hypothesis was corroborated by measurements of in vitro binding affinity showing that full-length StoD binds to both K48-linked and K63-linked diubiquitin with greater avidity than monoubiquitin.

StoD is not the only E3 ubiquitin ligase that has been shown to bind ubiquitin directly. HECT E3 ubiquitin ligases form a thioester bond between an internal cysteine residue and the C terminus of ubiquitin, which is essential for their activity (40) and a noncovalent interaction with ubiquitin is required for the activation of the RBR E3 ubiquitin ligase, Parkin (41). There are also more than 150 different UBDs (29), many of which have affinities for monoubiquitin of greater than 100 $\mu$M (42), whereas the concentration of ubiquitin within cells has been estimated to be 85 $\mu$M (43). This suggests that the binding affinities seen here for StoD towards both monoubiquitin and diubiquitin are within physiological limits and, furthermore, are directly comparable with those observed for other multivalent ubiquitin-binding proteins (31). The increased affinity of StoD for K63-linked diubiquitin over K48-linked diubiquitin appears to be due to the availability of the binding region on the surface of the ubiquitin molecules, suggesting that StoD may be involved in altering nondegradative cellular signaling pathways rather than those associated with proteasomal degradation. The "open" configuration seen within K63-linked diubiquitin is also observed in

linear diubiquitin, where the N terminus of one ubiquitin is connected to the C terminus of another by an isopeptide bond (29); therefore, future work may seek to assess the binding of linear diubiquitin to StoD.

The surface of StoD-N that interacts with ubiquitin coincides with the surface of NleG2-3 that is important for host protein recognition (13). Furthermore, ubiquitin binding by StoD-N is not a universal characteristic of the NleG family members, as NleG7-N and NleG8-N from *C. rodentium* did not bind ubiquitin in Y2H. As ubiquitin is present in free and conjugated forms throughout the host cell, our data could suggest that in contrast to targeting a specific substrate for ubiquitination, StoD globally recognizes and ubiquitinates pre-ubiquitinated targets. In this case, StoD would be a polyubiquitin "chain builder" rather than a "chain initiator," a discrimination more normally applied at the level of E2 ubiquitin–conjugating enzymes, where noncovalent interactions between the E2 and ubiquitin are also required to specifically drive elongation (44). Indeed, selective catalysis of multiubiquitin chain assembly is also the trademark of specialized U-box E3 ligase–denoted E4 enzymes, although these typically do not bind E2 enzymes and cooperate instead with a partner E3 enzyme (45). As StoD directly interacts with UBE2E1 and is capable of mediating autoubiquitination with a range of human E2 enzymes, it may represent a novel E4 enzyme. Thus, it is plausible that StoD can hijack host E2 enzymes to amplify ubiquitination pathways already present in the host cell. Alternatively, the striking localization of cellular ubiquitin into distinct puncta in the presence of overexpressed StoD could suggest that the effector subverts host cell pathways by concentrating or sequestering free or conjugated ubiquitin, although the in vivo effects of StoD at physiological levels still needs to be confirmed. Clearly, the implication of StoD-N ubiquitin binding on the physiological substrates of StoD requires further investigation.

In summary, this work identifies the first T3SS effector protein to be present in *S.* Typhi and not in *S.* Typhimurium and highlights the need to reassess the use of *S.* Typhimurium in the study of *S.* Typhi pathogenesis. Furthermore, the study revealed a novel class of bacterial E3 ligase effectors that can bind diubiquitin. A challenge for future work will be to identify the substrate(s) ubiquitinated by StoD and its role in *S.* Typhi infection.

# Materials and Methods

## Bioinformatics

The Kyoto Encyclopedia of Genes and Genomes (46) and National Center for Biotechnology Information website were used to retrieve sequences for sequence alignments performed using Clustal Omega (47) and formatted using Strap (48). The Maximum likelihood tree was based on the JTT matrix–based model (49) with 1,000-bootstrap replicates using MEGA7 (50).

## Bacterial strains and growth conditions

*Salmonella* strains (Table S1) were routinely cultured in LB Lennox (Sigma-Aldrich or Invitrogen) at 37°C, 200 rpm overnight. Where appropriate, antibiotics were used at the following concentrations: 30 $\mu$g/ml chloramphenicol (Cm$^R$), 50 $\mu$g/ml kanamycin (Kn$^R$), 100 $\mu$g/ml ampicillin (Amp$^R$), and 100 $\mu$g/ml spectinomycin (Spec$^R$). All antibiotics were purchased from Sigma-Aldrich. The *S.* Typhi mutants were generated using the $\lambda$ red recombinase system (51); the primers used are listed in Table S3.

## Plasmids

Plasmids used in this study are shown in Table S2. Genes were amplified from either *S.* Typhi (Ty2) or *C. rodentium* (ICC169) genomic DNA; their associated primers are listed in Tables S3 and S4. The gene sequence for *UBE2E1* was synthesized and subcloned between the *NdeI*/*EcoRI* sites of pET28b by Eurogentec Ltd. Mutagenic primers and the QuikChange XL Site-Directed Mutagenesis Kit (Agilent) were used to introduce point mutations in *ubiquitin*, *stoD*, and *nleG*.

## Tissue culture

HeLa cells (American Type Culture Collection [ATCC]) were cultured in DMEM containing 4,500 mg/l glucose (Sigma-Aldrich), supplemented with 10% (vol/vol) heat-inactivated FBS (Gibco), and 2 mM GlutaMAX (Gibco). THP-1 cells (ATCC) were cultured in suspension in Roswell Park Memorial Institute medium (RPMI-1640) containing L-glutamine (Sigma-Aldrich) supplemented with 10% (vol/vol) heat-inactivated FBS and 10 $\mu$M Hepes buffer (Sigma-Aldrich). Both cell lines were grown at 37°C and 5% $CO_2$ in a humidified environment and were regularly tested for mycoplasma using the MycoAlert Mycoplasma Detection Kit (Lonza). Cell invasion and intracellular replications assays were performed as described (7, 52).

## β-lactamase translocation assays

The $\beta$-lactamase translocation assay was performed as previously described (18). Briefly, HeLa cells, seeded in black-walled 96-well plates (BD Biosciences), were infected with *Salmonella* containing pWSK29-Spec (7) encoding TEM1-tagged effectors (Table S2) at a multiplicity of infection (MOI) of 100. Infected cells were centrifuged at 500 *g* for 5 min and incubated for 60 min at 37°C and 5% $CO_2$. The culture medium was replaced with 100 $\mu$l of 3 mM probenecid (Sigma-Aldrich), 20 mM Hepes in HBSS (Gibco), and 20 $\mu$l CCF2-AM LiveBLAzer-FRET B/G Loading Kit (Invitrogen) and incubated at room temperature, in the dark until 3 h postinfection. The cells were washed before the fluorescence was measured using a FLUOstar Optima plate reader (BMG Labtech) with an excitation wavelength of 410 nm and emission wavelengths of 450 and 520 nm. Response ratios were calculated by first subtracting the average background fluorescence for both 450 and 520 nm wavelengths from the fluorescence reading for each sample. The ratio of fluorescence at 450 nm to fluorescence at 520 nm for each sample was then divided by the uninfected ratio of fluorescence at these wavelengths.

## SPI-1 secretion assays

The SPI-1 secretion assay was performed as previously described (17). Briefly, overnight *Salmonella* cultures were diluted 1:33 into

50 ml LB and grown to an $OD_{600}$ of 1.8–2.0. 1 ml of the bacterial culture was pelleted and resuspended in 10 $\mu$l 2× SDS loading buffer per $OD_{600}$ of 0.1, for the expression sample. The remaining culture was cleared by centrifugation for 20 min at 4°C, 3,300 $g$ and the supernatant filtered through a 0.2 $\mu$m filter. Proteins were precipitated in 10% (vol/vol) trichloroacetic acid (Sigma-Aldrich), collected by centrifugation for 15 min, 20,000 $g$ at 4°C, and washed twice with ice-cold acetone. Protein pellets were air-dried before resuspension in 10 $\mu$l 2× SDS loading buffer per $OD_{600}$ of 0.1, giving the secreted sample. Both the expression and secreted samples were then boiled for 10 min at 100°C, before analysis by Western blot (8).

## Transfection and immunofluorescence staining

HeLa cells were seeded at $4.5 \times 10^4$ cells/well on coverslips in 24-well plates (BD Falcon) 24 h before transfection. The cell medium was replaced with fresh medium before transfection with eukaryotic expression vectors (Table S2). GeneJuice Transfection Reagent (Novagen) was used as per the manufacturer's instructions. Briefly, 0.75 $\mu$l GeneJuice Transfection Reagent was mixed with Opti-MEM containing GlutaMAX (Gibco) for 5 min before the addition of 0.25 $\mu$g DNA and incubation for 15–30 min at room temperature. For cotransfections, 0.25 $\mu$g of each vector was incubated with 1.5 $\mu$l of GeneJuice Transfection Reagent. This mixture was then added to the cells and incubated for 24 h at 37°C and 5% $CO_2$ in a humidified environment.

Transfected cells were fixed with 3.2% PFA (Agar Scientific) for 15–30 min. After washes, the cells were quenched in 50 mM ammonium chloride for 10 min before permeabilisation with 0.2% (vol/vol) Triton X-100 (Sigma-Aldrich). The cells were washed, blocked with 0.2% (wt/vol) BSA (Sigma-Aldrich), and incubated with primary antibodies (Table S5) diluted in 0.2% BSA in DPBS for 45–90 min. After washes, the coverslips were incubated with secondary antibodies (Table S5) and Alexa Fluor-647 Phalloidin (1:100 dilution; Invitrogen) or Phalloidin-iFluor 647 conjugate (1:1,000 dilution; Stratech) and Hoechst 33258 (1:1,000 dilution; Sigma-Aldrich) diluted in 0.2% BSA in DPBS, for 30 min. The coverslips were washed before mounting on microscope slides with Prolong Gold Antifade Reagent (Invitrogen). The stained cells were then viewed and analysed using Zeiss Axio Imager M1 or Zeiss Axio Observer Z1 (Carl Zeiss Microscopy) microscopes.

## Purification of recombinant StoD variants and *UBE2E1* for autoubiquitination assays

Cultures of *E. coli* BL21 (DE3) pLysS containing pET28a-*stoD* constructs (Table S2) were grown at 37°C to an $OD_{600}$ of 0.4–0.6. Protein expression was induced with 1 mM isopropyl $\beta$-D-1-thiogalactopyranoside (IPTG; Sigma-Aldrich) for 6 h at 30°C. Bacterial pellets were resuspended in His lysis buffer (20 mM Tris–HCl, pH 7.9, and 500 mM NaCl; Sigma-Aldrich) containing 1 mg/ml chicken egg white lysozyme (Sigma-Aldrich), 25 units benzonase nuclease (Novagen) per gram of bacterial pellet and cOmplete Mini EDTA-free protease inhibitor cocktail (Roche), lysed using an EmulsiFlex B15 cell disruptor (Avestin), and the soluble fraction was used to purify the StoD variants on a His-bind Resin (Novagen). UBE2E1 was either purchased from Ubiquigent or

produced in-house from *E. coli* BL21 Star cultures containing pET28b-*UBE2E1* (Table S2). Following centrifugation, UBE2E1 was purified from the soluble fraction by affinity chromatography using HiTrap TALON crude column (GE Healthcare Life Sciences). Protein fractions were analysed by SDS–PAGE and Coomassie stain. Selected protein fractions were then dialysed using SnakeSkin dialysis tubing (10 K molecular weight cutoff; Thermo Fisher Scientific) for 1–4 h and then again overnight in fresh His lysis buffer. Protein concentration was then determined using a NanoDrop 1000 (Thermo Fisher Scientific).

## E2 ubiquitin–conjugating enzyme screen

To assess which E2 ubiquitin–conjugating enzymes were capable of facilitating StoD autoubiquitination, the UbcH (E2) Enzyme Kit (Boston Biochem) containing UBE2K, UBE2H, UBE2R1, UBE2D1, UBE2D2, UBE2D3, UBE2E1, UBE2L3, UBE2E3, UBE2C, and UBE2N was used. The different E2s were used in combination with E1 ubiquitin-activating enzyme (Boston Biochem), biotinylated ubiquitin (Boston Biochem), 1,4-DTT (Sigma-Aldrich) and buffered ATP solution (Boston Biochem) according to the manufacturer's instructions. Reactions were then boiled for 5 min at 100°C before analysis by Western blotting.

## Autoubiquitination assays

StoD autoubiquitination assays were performed using a protocol adapted from the E2 Scan Kit (Ubiquigent) (53). Variants of StoD were incubated at a concentration of 1 $\mu$M with 0.1 $\mu$M $His_6$-UBE1 (Ubiquigent), 100 $\mu$M ubiquitin (Ubiquigent), 0.05 nmoles $His_6$-UBE2E1 in 50 mM Hepes, pH 7.5, 5 mM $MgCl_2$ (Sigma-Aldrich), and 5 mM DTT (Sigma-Aldrich) with or without 2 mM ATP (Thermo Fisher Scientific) for 1 h at 30°C. The reaction was then stopped by adding 50% (vol/vol) glycerol, 0.3 M Tris–HCL, pH 6.8 (Sigma-Aldrich), 10% (wt/vol) SDS (Merck), 5% (vol/vol) $\beta$-mercaptoethanol (Sigma-Aldrich), and 0.05% (wt/vol) bromophenol blue (5× SDS loading buffer) and boiled for 5 min at 100°C before analysis by Western blot.

## Direct Y2H assays

Y2H was performed as previously described (54). Briefly, *Saccharomyces cerevisiae* AH109 was cotransformed with 500 ng of both pGBKT7-*bait* and pGADT7-*prey* vectors (Table S2) and plated onto SD agar plates lacking L-leucine and L-tryptophan (double dropout [DDO]) to select for cotransformants. Transformants were allowed to grow for 3 d at 30°C before being resuspended in sterile water and spotted onto DDO and QDO (quadruple dropout lacking Trp, Ade, His, and Leu and supplemented with 40 $\mu$g/ml X-$\alpha$-gal) plates to assess the interaction of bait and prey proteins.

## Protein purification for crystallisation

StoD [1–101] with an N-terminal MGSSHHHHHHSSGLVPRGSH tag (Table S2) was expressed and purified as for autoubiquitination assays, except expression was induced for 16 h at 21°C and the His-tagged protein extracted using a 5 ml $Ni^{2+}$-NTA superflow cartridge

(QIAGEN). The eluate was directly applied to a HiLoad 16/60 Superdex 75 pg (GE Healthcare) column equilibrated in 20 mM Tris–HCl, pH 7.5, and 150 mM NaCl. The protein was concentrated in a centrifugal concentrator device (10 kD molecular mass cutoff membrane; Millipore) to 20 mg/ml. Selenomethionine (SeMet)-substituted StoD [1–101] was expressed in B834 (DE3) cells using SelenoMethionine Medium Complete (Molecular Dimensions) and then purified in the same way.

Protein was crystallised at 21°C by the vapour diffusion sitting-drop method with 400 nl drops using an OryxNano Crystallisation Robot (Douglas Instruments). Native crystals grew with 60% 0.9 M Na malonate, 0.5% Jeffamine, 0.1 M Hepes, pH 6.5, whereas SeMet crystals grew with 50% 0.9 M Na malonate, 0.5% Jeffamine, and Hepes, pH 6.9.

### Data collection, structure determination, and refinement

Diffraction data were collected at the European Synchrotron Radiation Facility (ESRF) (Beamline ID29) at 120K from one native and one SeMet-labelled crystal ($\lambda$ = 0.972). The data were processed using the Xia2 (55) pipeline in the 3da mode. Eight selenium sites, phases, and an initial solvent-flattened electron density map were calculated from the SeMet dataset using autoSHARP (56). The output was combined with the native dataset using CAD to produce an improved electron density map. Buccaneer (57) was subsequently able to autobuild a model with 410 residues with four copies in the asymmetric unit. Further rounds of building using COOT (58) and refinement in autoBUSTER (59) were carried out to give a final model with 429 residues. Protein chemistry was validated with MolProbity (60) and the final model visualised with PyMol (Schrödinger).

### Protein purification for NMR spectroscopy, microscale thermophoresis, and MALS

Unlabelled StoD variants were purified as for crystallisation, except the Ni$^{2+}$ eluate was dialysed overnight at 4°C with thrombin (Amersham Biosciences) to remove the His tag before size exclusion chromatography (SEC). UBE2E1 was purified as for autoubiquitination assays, except the Ni$^{2+}$ eluate was dialysed overnight at 4°C with thrombin to remove the His tag and subsequently applied to a HiLoad 26/60 Superdex 75 pg (GE Healthcare) equilibrated in 20 mM Tris–HCl, pH 7.5, 150 mM NaCl, and 1 mM TCEP.

Human ubiquitin variants were expressed in BL21 (DE3) cells cotransformed with pET3a-*ubiquitin* and pJY2 constructs (Table S2) as previously described (61). The cells were resuspended in 50 mM Tris–HCl, pH 7.6, 10 mM MgCl$_2$, 0.02% Triton X-100, and 0.1 mg/ml DNase with a Protease Inhibitor Tablet (Pierce) and lysed using an Emulsiflex-C5 Homogeniser (GC Technologies). 1% (vol/vol) perchloric acid was added dropwise to the clarified lysate on ice and stirred for 30–45 min. After removal of the precipitate by centrifugation, 5 M NaOH was added to reach pH 8, and the solution dialysed overnight against 50 mM Tris–HCl, pH 7.5, using SnakeSkin dialysis tubing (3.5 K molecular weight cutoff; Thermo Fisher Scientific). The protein was concentrated in a centrifugal concentrator device (3 kD molecular mass cutoff membrane; Millipore) and applied to a HiLoad 26/60 Superdex 75 pg (GE Healthcare) equilibrated in 20 mM Tris–HCl, pH 7.5, 150 mM NaCl, and 2 mM TCEP.

Isotope-labelled StoD-N [1–101], StoD-C [134–233] and ubiquitin were expressed in $^{15}$N ($\pm^{13}$C)-labelled M9 minimal medium and purified as for unlabelled protein. SEC was performed in 20 mM Tris–HCl, pH 7.5, 150 mM NaCl (supplemented with 1 mM TCEP for StoD-C [134–233]) for $^{15}$N-labelled proteins, and 25 mM NaPi, pH 7.0, for $^{13}$C/$^{15}$N–labelled proteins.

### NMR spectroscopy

5% (vol/vol) D$_2$O was added to all samples. All spectra were recorded at 298K on a Bruker Avance II 500 MHz Spectrometer. Backbone $^1$H, $^{15}$N, and $^{13}$C assignments of $^{13}$C/$^{15}$N–labelled 545 $\mu$M StoD-N [1–101] and 575 $\mu$M StoD-C [134–233] were achieved using CBCA(CO)NH (62) and CBCANH (63) experiments. Backbone $^1$H and $^{15}$N assignments for human ubiquitin were obtained from BMRB entries 68 (64) and 2,573 (65), respectively. NMR titrations with various ligands were performed by collecting $^1$H, $^{15}$N-HSQC spectra of $^{15}$N–labelled proteins at 100 $\mu$M. Spectra were processed using TopSpin (Bruker) and analysed with Sparky (66).

### Diubiquitin synthesis

Ubiquitin variants were concentrated to 4 mM. K48-linked diubiquitin was synthesized in 1 ml of 50 mM Tris–HCl, pH 8.0, 2 mM TCEP supplemented with 1× energy regeneration solution (BostonBiochem), 100 nM His$_6$-Ube1 (BostonBiochem), 2.5 $\mu$M E2-25K (BostonBiochem), 1 mM ubiquitin$_{G76C}$, and 1 mM ubiquitin$_{K48R}$. K63-linked diubiquitin was synthesized in the same buffer supplemented with 1× energy regeneration solution, 100 nM His$_6$-Ube1, 2.5 $\mu$M His$_6$-UBE2N/Uev1a complex (BostonBiochem), 1 mM ubiquitin$_{G76C}$, and 1 mM ubiquitin$_{K63R}$. Reactions were incubated at 30°C for 16 h then flowed through a 1 ml Ni$^{2+}$-NTA superflow cartridge (QIAGEN) to extract the His$_6$-tagged E1/E2 enzymes. Unreacted ubiquitin and diubiquitin were then separated on a HiLoad 26/60 Superdex 75 pg (GE Healthcare) equilibrated in 20 mM Hepes, pH 7.5, 150 mM NaCl, and 2 mM TCEP.

### Microscale thermophoresis

All proteins were dialysed into 20 mM Hepes, pH 7.5, 150 mM NaCl, 2 mM TCEP, and 0.02% Tween. The lysine residues of StoD variants were labelled using the RED-NHS Labeling Kit (NanoTemper Technologies), whereas the single cysteine of ubiquitin$_{G76C}$ variants was labelled using the RED-MALEIMIDE Labeling Kit (NanoTemper Technologies), both according to the manufacturer's instructions. One in two dilution series of ubiquitin in the range of 61 nM–2 mM or StoD in the range of 16 nM–500 $\mu$M were mixed with 40 nM labelled protein. Thermophoresis was measured using a Monolith NT.115 instrument (NanoTemper Technologies) at 22°C using standard treated capillaries (NanoTemper Technologies). For titration of the labelled StoD variants with ubiquitin, data were analysed using the signal from thermophoresis ± T jump (NT Analysis software version 1.5.41; NanoTemper Technologies). For titration of labelled ubiquitin variants with StoD, the capillary scan in the NT Analysis software at 40% LED power already showed concentration-dependent fluorescence changes. Denaturation of these mixtures and re-measurement of their fluorescence in an SD test (67) confirmed these fluorescence

changes were due to ligand binding, allowing fluorescence values to be used directly for $K_D$ determination.

### SEC-MALS

SEC was performed with a Superdex 75 10/300 (GE Healthcare) equilibrated in 20 mM Tris–HCl, pH 7.5, and 150 mM NaCl. 100 $\mu$l protein was injected at increasing concentrations. The column was followed in-line by a Dawn Heleos-II light scattering detector (Wyatt Technologies). Molecular weight calculations were performed using ASTRA 6.1.1.17 software (Wyatt Technologies) assuming a dn/dc value of 0.186 ml/g.

### Statistical analysis

All data were analysed using GraphPad Prism 7 (GraphPad Software).

### Accession codes

The coordinates and structure factors for StoD-N have been deposited in the RCSB PDB with ID code 6IAI.

# Supplementary Information

# Acknowledgements

We thank Del Pickard and Gordon Dougan from the Wellcome Trust Sanger Institute for providing strains and technical help and Vassilis Koronakis from the University of Cambridge for providing anti-effector antibodies. We thank Mariella Lomma, Cedric N. Berger, and Ranjani Ganji for their experimental and intellectual contributions. This project was supported by a Wellcome Trust Investigator Award 100298/Z/12/Z (S Lea), a Wellcome Trust Investigator Award 107057/Z/15/Z (G Frankel), and BBSRC grant BB/K001515/1 (G Frankel). AMP Byrne, R Johnson, E Mylona, and A Sagfors were supported by PhD studentships from the BBSRC and the MRC.

## Author Contributions

MA McDowell: data curation, formal analysis, supervision, investigation, methodology, writing—original draft, review, and editing.
AMP Byrne: investigation and writing—original draft, review, and editing.
E Mylona: data curation, investigation, methodology, and writing—review and editing.
R Johnson: data curation, investigation, methodology, and writing—original draft.
A Sagfors: investigation.
VF Crepin: supervision.
S Lea: data curation, formal analysis, supervision, funding acquisition, and writing—original draft.
G Frankel: supervision, funding acquisition, project administration, and writing—original draft, review, and editing.

## Conflict of Interest Statement

The authors declare that they have no conflict of interest.

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
