## [Reviewer comments · Life Science Alliance]

Life Science Alliance

The *S. Typhi* effector StoD is an E3 ubiquitin ligase which binds K48- and K63-linked di-ubiquitin

Melanie McDowell, Alexander Byrne, Elli Mylona, Rebecca Johnson, Agnes Sagfors, Valerie Crepin, Susan Lea, and Gad Frankel

DOI: <https://doi.org/10.26508/lsa.201800272>

Corresponding author(s): Gad Frankel, Imperial College

Review Timeline:

Submission Date:	2018-12-10
Editorial Decision:	2019-01-28
Revision Received:	2019-04-10
Editorial Decision:	2019-04-30
Revision Received:	2019-05-03
Accepted:	2019-05-07

Scientific Editor: Andrea Leibfried

Transaction Report:

January 28, 2019

Re: Life Science Alliance manuscript #LSA-2018-00272-T

Prof. Gad Frankel
Imperial College
CMMI
EXHIBITION ROAD
London SW7 2AZ
United Kingdom

Dear Dr. Frankel,

Thank you for submitting your manuscript entitled "The S. Typhi effector StoD is an E3 ubiquitin ligase which binds K48- and K63-linked di-ubiquitin" to Life Science Alliance. The manuscript was assessed by expert reviewers, whose comments are appended to this letter.

As you will see, the three reviewers all think that there is value in your findings. However, they also think that the conclusions put forward need to be backed up with additional support. We would thus like to invite you to submit a revised version of your work, addressing the concerns raised by the reviewers. It is not mandatory for publication here to provide further biological insight (point 2 of ref#1; second point of ref#2), however all other points raised by the reviewers need to get addressed.

Thank you for this interesting contribution to Life Science Alliance. We are looking forward to receiving your revised manuscript.

Sincerely,

- A letter addressing the reviewers' comments point by point.
- An editable version of the final text (.DOC or .DOCX) is needed for copyediting (no PDFs).
- High-resolution figure, supplementary figure and video files uploaded as individual files: See our detailed guidelines for preparing your production-ready images, <http://life-science-alliance.org/authorguide>
- Summary blurb (enter in submission system): A short text summarizing in a single sentence the study (max. 200 characters including spaces). This text is used in conjunction with the titles of papers, hence should be informative and complementary to the title and running title. It should describe the context and significance of the findings for a general readership; it should be written in the present tense and refer to the work in the third person. Author names should not be mentioned.

B. MANUSCRIPT ORGANIZATION AND FORMATTING:

Full guidelines are available on our Instructions for Authors page, <http://life-science-alliance.org/authorguide>

Reviewer #1 (Comments to the Authors (Required)):

The manuscript by McDowell et al describes experiments that characterize the *S. typhi* effector StoD. This is the first report of an *S. typhi* effector and it turns out to be a homologue of EPEC NleG.

StoD is of interest because so much of *S. typhi* pathogenesis is not understood and the observation that *S. typhi* NleG homologues are evolutionarily different from the ancestors. Furthermore, this work points out that studies of *S. typhimurium* are not actually a good model for *typhi* and this needs to be emphasized in the pathogenesis community.

The authors show by standard assays that StoD is secreted via a SPI mechanism and that autoubiquitination activity resides in the C-term of StoD (confirmed by NMR chemical shift changes) and requires a conserved proline residue P204. Together, along with homology modeling and mutant analysis, the data convincingly show that StoD-C has a canonical U-box E3 ligase domain. The crystal structure of the N-terminus was solved (since it has low homology to other homologues) and was shown to have a ubiquitin-like fold. In vitro and in vivo imaging analyses indicated that StoD binds ubiquitin. Interestingly, both domains seem to bind ubiquitin.

1. It would be good to see the in vivo imaging experiment performed with the N and C-terminal constructs of StoD. Are similar puncta observed?

2. The unanswered question from this work is what are the host consequences of StoD being present in the cytoplasm? The authors convincingly show that StoD binds and is co-localized with host ubiquitin, but what does this mean to the host?

Minor points:

Abstract: the sentence In HeLa cells.....is confusing. Distribution should be distributed? Last sentence, please add additional sentence regarding chain building. What would be the host or bacterial significance?

Lines 186-187 please explain why this is striking for StoD to have a ubiquitin like domain and E3 ligase function for readers who are outsiders.

How does a K_d of 5-15 μM compare with other ubiquitin binding proteins? This is discussed in the discussion, but a brief sentence would be useful in the results. What were the Hill coefficients in the binding assays shown in Fig. 5?

Line 215 of should be on

Reviewer #2 (Comments to the Authors (Required)):

The manuscript by McDowell and colleagues focuses on the molecular function and structure of type 3 secreted effector StoD from *S. Typhi*. This pathogenic factor belongs to a large family of homologous functionally diverse NleG effectors present in *E. coli* and *Salmonella* strains. The most conserved C-terminal portion of NleG effectors represents a functional U-box domain able to engage a specific set of host ubiquitination machinery members.

Through the combination of structural, biophysical and cellular assays McDowell and colleagues demonstrate that StoD is composed of distinct N- and C- terminal domains which are able to bind human ubiquitin. Based on this data authors formulate a hypothesis by which StoD facilitates the formation of polyubiquitinated targets in the host cell.

The findings presented in this manuscript advance our understanding of NleG family of pathogenic

factors that play an important role in the infection strategy of important human and animal bacterial pathogens. Based on this I would recommend this manuscript for publication given that the points I raise below are adequately addressed.

General points to address:

- Many acronyms are not described before use or at all, and there are several typos. Please fix throughout the manuscript.
- Does StoD colocalize with Salmonella during infection as ubiquitin does during infection?
- Why do the authors switch from L167K at the beginning to disrupt E2 interaction, but then use P204K in later work? Please be consistent throughout. In the Y2H assays, the authors suggest that P204K impairs interaction with ubiquitin directly despite this being an E2- interaction residue. Additionally, the Y2H interaction with ubiquitin could be mediated by an E2. Please test in vitro if the P204K and L167K mutation can still bind to ubiquitin directly.
- Lines 330-332: To adequately make this claim that the StoD targeting is pre-ubiquitinated proteins, the authors should confirm that other NleGs do not also possess the ability to bind ubiquitin in their N-terminus. If other NleGs can recognize ubiquitin as well, this suggests that StoD would have an additional N-terminus host target, in addition to ubiquitin. Perhaps using NleG2-3 and its host recognition target, because as the authors say, NleG2-3 has a verified target that is recognized its target through a similar interface. This is an essential experiment.

Specific notes:

Line 114+116: Reference for Δ prgH Δ ssaV causing deficiency in secretion missing.

Fig 2C. The authors don't state how they aligned the structures/what the length of alignment is. Would be good to show comparisons to the published structures in the E2/E3 complex directly instead of just a model.

Fig S5C legend: Refers to the model in "Fig 3C" - there is no fig 3C, but presumably this refers to the model from fig 2C. Assuming this is correct, the authors should fix this.

Line 165: the SEC-MALS acronym is not described anywhere in manuscript. Please fix.

Fig. S8 - This figure is difficult to interpret and misleading. Please show the whole chromatogram - one elution peak is not sufficient to determine if the protein is monomeric - this is essential to include. Please describe what the horizontal dotted line across the peak corresponds to in terms of the MALS output and size calculations. If the N and C-terminal domains are monomeric, what about the full length? It also would be good to include legend in figure itself corresponding to the colours of the different protein concentrations.

Fig S9A - It would be good to show the structure surface for visualizing B-factor - cartoon makes it difficult to see loop b fac.

Line 172 - says NleG 5.1 instead of NleG5-1 elsewhere. This happens a few times throughout SI as well, please fix.

Fig 3B - its hard to visualize the overlap between the N-terminal domain and ubiquitin - consider adding another perspective for this superimposition

Lines 190: Given that StoD can be secreted as shown in Fig. 1, is there a explanation for why it can't be detected during infection? Perhaps the authors would consider using a different antibody for immunofluorescence?

Line 195: Please add this negative Rab11, Vamp3, LC3 data to the supplemental as or colocalization tests with ubiquitin-like proteins.

Line 214-216: This could also suggest that the C-terminus is dependent on E2 interaction and subsequent autoubiquitination to bind to ubiquitin in Y2H. This could potentially be tested by mutating the C-terminal G76 of ubiquitin in this Y2H assay. Also, if the P204K mutation is indeed making the E3 ligase domain fold incorrectly, please also test this with the L167K mutation used earlier.

Fig S12. Should ideally use the same scale bar format as in for the MST in Fig. 5.

For Di-ubiquitin synthesis, a coomassie gel would be helpful to show that only Di-ubiquitin variants were formed as expected.

Line 853: typo - "duef to line", change to "due to line"

Methods issues.

-In general, the methods are missing several a few key points - please go through thoroughly and fill in gaps.

-In supplemental materials, all tables are formatted to say "Table S1" or other errors.

-For the Y2H assays, please indicate why the strains representing a positive interaction are blue, and explain in better terms how the assay is performed - for example, what are QDO plates? What does DDO stand for? What are the additional components missing from QDO/ presumably there is b-gal in the media.

-Line 570: No where throughout paper is MALS defined. There could be more detail here. Also, presumably this section should be titled SEC-MALS?

-Line 459: remove "An" from method title, should read "E2 ubiquitin-conjugating enzyme screen" or similar.

Reviewer #3 (Comments to the Authors (Required)):

This manuscript by McDowell et al. describes a novel, *S. Typhi*-specific effector E3 ligase with a peculiar ability to bind Ub chains. As the authors posit, this could indicate that StoD is involved in extending existing Ub chains within the host cell, resulting in local amplification of signals that would be consistent with the Ub-stained puncta observed following StoD overexpression. The authors use rigorous structural and biophysical analyses to characterize E2 and Ub binding to the StoD U-box and ubiquitin-like domains. The study provides fascinating insights that will be appreciated by bacteriologists, structural biologists, and ubiquitin biochemists. The work is well done but lacks several controls and biochemical experiments that could help link their observations to biological function. Without these, the work showing ligase function, Ub puncta formation, and Ub chain binding are very disjointed and, on their own, less exciting. My recommendation would be for the authors to revise and resubmit.

Main concerns:

1) The nature of ubiquitination observed in Figure 2 is unclear without additional Western blots and controls. If the authors would like to present this as auto-ubiquitination, an anti-StoD blot is needed. Regardless, the authors should do a 'No-E3' control, as many of the E2s shown in their panel have known E3-independent activity that could be confounding their analysis. Particularly, while the strong Ub-modification at around 38 kDa in Fig. 2A is most likely mono-ubiquitinated StoD, the banding patterns for UBE2K and UBE2H are more consistent with E3-independent activity. Further, if the ubiquitination observed in these blots is, in fact, auto-ubiquitination, why does the banding pattern not shift in Fig. 2B as the molecular weight of StoD changes?

2) The biochemical analysis of the U-box-E2 interface should be more complete. Ideally, NMR binding and in vitro activity assays should be shown for both the P204K and L167A mutants. The sites of these mutations should also be mapped onto the model in Fig. 2C. As an explanation for their Y2H data, the authors propose that the P204K mutant might be misfolded; showing this by NMR should be straightforward and informative.

3) An effort should be made to separate E2 and Ub binding to the StoD U-box by point mutation. The authors use the P204K mutation for auto-ubiquitination and cellular overexpression analyses, but the effects of this mutation on the two binding interfaces are unclear. Using their NMR binding analyses, the authors should be able to identify interface-specific mutations that can be used for clarifying their biochemical and cellular phenotypes. Likewise, identifying a Ub interface mutant in the ubiquitin-like domain would provide important insight into the role this plays in auto-ubiquitination activity and cellular puncta formation.

4) The authors should make more of an effort to place their studies on StoD into the context of other NleG family effectors. Would they predict other examples to bind Ub at the U-box domain? And at the ubiquitin-like domain?

Minor concerns:

1) Consider changing the 'chain builder' nomenclature to be more consistent with the literature, which tends to describe chain extending E3 ligases as E4 enzymes.

2) The broken Western blots, particularly in Fig. 1C, make comparisons very difficult. If possible, please provide side-by-side blots or at least full-gel blots in the Supplemental.

3) The authors reference numerous immunofluorescence experiments as 'data not shown'. Although this is negative data, it should still be included in the Supplemental if it is to be mentioned at all.

4) I'd like to point out that the authors repeatedly stress the importance of not using *S. Typhimurium* as a surrogate for studying *S. Typhi*, but then proceed to do this to test StoD secretion and its role (or lack thereof) in infection. I understand this was a necessary swap, but perhaps the authors should provide further explanation for why this work is still valid.

5) While the text is well-written overall, I would suggest an additional read through looking at comma usage.

6) Please provide a PDB accession code for your structure deposition.

Reviewer #1:

Comment 1: It would be good to see the in vivo imaging experiment performed with the N and C-terminal constructs of StoD. Are similar puncta observed?

Response: We have now performed these experiments. Transfection of StoD-N caused re-distribution of ubiquitin in 5.5% of transfected cells while transfection of StoD-C did not cause re-distribution of ubiquitin. These results are now presented in Fig. 4B and 4C.

Comment 2: How does a Kd of 5-15 uM compare with other ubiquitin binding proteins? This is discussed in the discussion, but a brief sentence would be useful in the results. What were the Hill coefficients in the binding assays shown in Fig. 5?

Response: Comment added to results and discussion section about Kd values. Hill coefficients added to figure legend.

Reviewer #2

Comment 1: Why do the authors switch from L167K at the beginning to disrupt E2 interaction, but then use P204K in later work? Please be consistent throughout.

Response: We added new data, Fig. 2A, showing that alanine substitution of L167 did not affect the E3 ligase activity of StoD, as reported for NleG3-2, hence the switch to P204K for the autoubiquitination assay, in line with Wu et al. 2010.

For NMR experiments, mild alanine mutations in the E2 binding site were initially screened to identify those which caused minimal disruption to the structure of StoD-C, a criteria unlikely to have been met by P204K. Importantly, L167A did not change the fold of StoD-C and yet severely disrupted binding to UBE2E1, providing strong evidence that this residue is within the E2 interaction surface. As autoubiquitination assays were performed with an excess of StoD, the L167A mutant may still undergo a weak interaction with UBE2E1 that is sufficient for autoubiquitination but is not observable in the equimolar NMR titration. A comment has been added to the results section to explain this apparent discrepancy.

Comment 2: In the Y2H assays, the authors suggest that P204K impairs interaction with ubiquitin directly despite this being an E2- interaction residue. Additionally, the Y2H interaction with ubiquitin could be mediated by an E2. Please test in vitro if the P204K and L167K mutation can still bind to ubiquitin directly.

Response: We agree it is strange P204K no longer interacts with ubiquitin as it is not within the ubiquitin binding surface of StoD-C suggested by NMR. This indicates P204K is having a much larger effect on the global fold of StoD-C.

Importantly, NMR titration of StoD-C with ubiquitin and UBE2E1 together suggests that the ternary complex forms. In addition, NMR data shows that UBE2E1 and ubiquitin do not interact directly at these concentrations (now added as Fig. S19B) so StoD-C is interacting directly with both ubiquitin and UBE2E1. We therefore present strong evidence that in vitro the interaction of StoD-C with ubiquitin is not via the E2.

Comment 2: Lines 330-332: To adequately make this claim that the StoD targeting is pre-ubiquitinated proteins, the authors should confirm that other NleGs do not also possess the ability to bind ubiquitin in their N-terminus. If other NleGs can recognize ubiquitin as well, this suggests that StoD would have an additional N-terminus host target, in addition to ubiquitin.

Perhaps using NleG2-3 and its host recognition target, because as the authors say, NleG2-3 has a verified target that is recognized its target through a similar interface. This is an essential experiment.

Response: We added new data (Fig. S15) showing that in NleGs from *Citrobacter rodentium* (see Fig. 1), full length NleG7 and NleG8, as well as NleG7-C, but not NleG7-N or NleG8-N bind ubiquitin (NleG8-C was toxic in the yeast and could not be tested).

Comment 3: Fig 2C. The authors don't state how they aligned the structures/what the length of alignment is. Would be good to show comparisons to the published structures in the E2/E3 complex directly instead of just a model.

Response: Superimpositions were done using Superpose. The lengths of all alignments are already given in the corresponding figure legends together with the RMSD. Fig S6C is now an overlay of the model with the CHIP/UBE2E1 structure.

Comment 4: Line 165: the SEC-MALS acronym is not described anywhere in manuscript.

Response: Done

Comment 5: Fig. S8 - This figure is difficult to interpret and misleading. Please show the whole chromatogram - one elution peak is not sufficient to determine if the protein is monomeric - this is essential to include. Please describe what the horizontal dotted line across the peak corresponds to in terms of the MALS output and size calculations. If the N and C-terminal domains are monomeric, what about the full length? It also would be good to include legend in figure itself corresponding to the colours of the different protein concentrations.

Response: The figure was modified to include the entire chromatogram, MALS for the full-length protein and in-figure legends. Figure legend and material & methods describe output as is conventional for SEC-MALS.

Comment 6: Fig S9A - It would be good to show the structure surface for visualizing B-factor - cartoon makes it difficult to see loop b fac.

Response: We believe this would be inappropriate – B factors are conventionally shown for the backbone C α only and thus can only be mapped on a backbone representation of the structure i.e. cartoon. Mapping on the surface representation would imply we are also commenting on the side chain B factors.

Comment 7: Fig 3B - its hard to visualize the overlap between the N-terminal domain and ubiquitin - consider adding another perspective for this superimposition

Response: Done

Comment 8: Lines 190: Given that StoD can be secreted as shown in Fig. 1, is there a explanation for why it can't be detected during infection? Perhaps the authors would consider using a different antibody for immunofluorescence?

Response: We don't have an explanation at this stage why StoD is not detected by infection, particularly as the HA antibodies detect transfected StoD. We are now addressing this in the discussion.

Comment 9: Line 195: Please add this negative Rab11, Vamp3, LC3 data to the supplemental as or colocalization tests with ubiquitin-like proteins.

Response: Done

Comment 10: Line 214-216: This could also suggest that the C-terminus is dependent on E2 interaction and subsequent autoubiquitination to bind to ubiquitin in Y2H. This could potentially be tested by mutating the C-terminal G76 of ubiquitin in this Y2H assay. Also, if the P204K mutation is indeed making the E3 ligase domain fold incorrectly, please also test this with the L167K mutation used earlier.

Response: We accept these are good suggestions, but we believe this is now covered by the new data listed above. Performing the suggested experiments would take us beyond the 3 months revision period and would form part of the follow-on studies.

Comment 11: Fig S12. Should ideally use the same scale bar format as in for the MST in Fig. 5.

Response: The legend was updated to show graphs in same representation as Fig. 5 and 6.

Comment 12: For Di-ubiquitin synthesis, a coomassie gel would be helpful to show that only Di-ubiquitin variants were formed as expected.

Response: Fig. S22 now inserted to show di-ubiquitin SEC profiles and gels.

Methods issues.

Comment 13: In supplemental materials, all tables are formatted to say "Table S1" or other errors.

Response: Done

Comment 14: For the Y2H assays, please indicate why the strains representing a positive interaction are blue, and explain in better terms how the assay is performed - for example, what are QDO plates? What does DDO stand for? What are the additional components missing from QDO/ presumably there is b-gal in the media.

Response: The information has been added to the relevant section in material and methods.

Comment 15: Line 570: Nowhere throughout paper is MALS defined. There could be more detail here. Also, presumably this section should be titled SEC-MALS?

Response: Done

Reviewer #3

Comment 1: The nature of ubiquitination observed in Figure 2 is unclear without additional Western blots and controls. If the authors would like to present this as auto-ubiquitination, an anti-StoD blot is needed.

Response: We have replaced Fig. 2A with a new WB showing the ubiquitination activity of WT and L167 StoD. The bottom panel, anti His, now shows a ladder of poly ubiquitinated StoD.

Comment 2: Regardless, the authors should do a 'No-E3' control, as many of the E2s shown in their panel have known E3-independent activity that could be confounding their analysis. Particularly, while the strong Ub-modification at around 38 kDa in Fig. 2A is most likely mono-ubiquitinated StoD, the banding patterns for UBE2K and UBE2H are more consistent with E3-independent activity. Further, if the ubiquitination observed in these blots is, in fact, auto-ubiquitination, why does the banding pattern not shift in Fig. 2B as the molecular weight of StoD changes?

Response: This is a good point. We have tested multiple E2 as a screen for selection of a suitable partner for the structural studies. For this reason, the original Fig. 2A has been moved to the supplementary. Moreover, although we don't have a 'No-E3' control in Fig. 2B, the fact that no signal is seen in the presence of E2 and StoD_{P204K} suggest there is no E3-independent activity.

Comment 3: The biochemical analysis of the U-box-E2 interface should be more complete. Ideally, NMR binding and in vitro activity assays should be shown for both the P204K and L167A mutants. The sites of these mutations should also be mapped onto the model in Fig. 2C. As an explanation for their Y2H data, the authors propose that the P204K mutant might be misfolded; showing this by NMR should be straightforward and informative.

Response: Fig. 2 now shows in vitro activity assay for both the P204K and L167A mutants. P204 now shown on Fig. 2C. L167 already shown clearly in Fig. S6D. P204K was not used for NMR because mild alanine mutations in the E2 binding site were already identified that caused minimal disruption to the structure of StoD-C and still disrupted the E2 interaction.

Comment 4: An effort should be made to separate E2 and Ub binding to the StoD U-box by point mutation. The authors use the P204K mutation for auto-ubiquitination and cellular overexpression analyses, but the effects of this mutation on the two binding interfaces are unclear. Using their NMR binding analyses, the authors should be able to identify interface-specific mutations that can be used for clarifying their biochemical and cellular phenotypes. Likewise, identifying a Ub interface mutant in the ubiquitin-like domain would provide important insight into the role this plays in auto-ubiquitination activity and cellular puncta formation.

Response: These are indeed interesting experiments to perform, but we are unable to perform these experiments for this revision. These studies would form the basis for our follow-on studies and a grant application.

Comment 5: The authors should make more of an effort to place their studies on StoD into the context of other NleG family effectors. Would they predict other examples to bind Ub at the U-box domain? And at the ubiquitin-like domain?

Response: See response to comment 2 by reviewer 2.

Minor concerns:

Comment 6: Consider changing the 'chain builder' nomenclature to be more consistent with the literature, which tends to describe chain extending E3 ligases as E4 enzymes.

Response: We added few sentences into the discussion about this, but designate StoD as a novel E4 enzyme due to its direct interaction with E2 enzymes.

Comment 7: The broken Western blots, particularly in Fig. 1C, make comparisons very

difficult. If possible, please provide side-by-side blots or at least full-gel blots in the Supplemental.

Response: We have replaced Fig. 1C as the secretion assays were repeated in *S. Typhimurium*. Secretion assay with *S. typhi* are notoriously difficult (one has to use a non-capsulated strain), particularly as they have to be done in a Cat3 lab.

Comment 8: The authors reference numerous immunofluorescence experiments as 'data not shown'. Although this is negative data, it should still be included in the Supplemental if it is to be mentioned at all.

Response: This has been added as Fig. S12, S13.

Comment 9: I'd like to point out that the authors repeatedly stress the importance of not using *S. Typhimurium* as a surrogate for studying *S. Typhi*, but then proceed to do this to test StoD secretion and its role (or lack there of) in infection. I understand this was a necessary swap, but perhaps the authors should provide further explanation for why this work is still valid.

Response: Only secretion assays were done in the *S. Typhimurium* background (due to safety concerns); all the functional assays were done in *S. Typhi* (Fig. S2).

Comment 10: Please provide a PDB accession code for your structure deposition.

Response: Done. The structure is deposited with PDB ID 6IAI.

April 30, 2019

RE: Life Science Alliance Manuscript #LSA-2018-00272-TR

Prof. Gad Frankel
Imperial College
CMMI
EXHIBITION ROAD
London SW7 2AZ
United Kingdom

Dear Dr. Frankel,

Thank you for submitting your revised manuscript entitled "The S. Typhi effector StoD is an E3 ubiquitin ligase which binds K48- and K63-linked di-ubiquitin". As you will see, the reviewers appreciate the introduced changes. However, a few concerns remain, which should get addressed by text changes (see comments of reviewer #3) prior to acceptance of your paper. Please also include the following in the final revision:

- please list 10 authors et al. in the reference list
- note that the legend of Figure S13 does not mention the sub-panels, please fix
- we display S figures in-line in the HTML version of the paper, these have to be provided as single-page files therefore. Please fix Figure S1 (currently runs over two pages)
- please upload all figures, including S figures, as individual files
- uploaded tables S2 and S3 may have wrong headers ('error' and 'S1'), please check and fix
- please link your ORCID iD to your profile in our submission system, you should have received an email with instructions on how to do so
- note that we follow ICMJE authorship guidelines, please check the author contributions and adapt accordingly (<http://www.icmje.org/recommendations/browse/roles-and-responsibilities/defining-the-role-of-authors-and-contributors.html>)

A. FINAL FILES:

B. MANUSCRIPT ORGANIZATION AND FORMATTING:

Sincerely,

Andrea Leibfried, PhD
Executive Editor
Life Science Alliance
Meyerhofstr. 1
69117 Heidelberg, Germany

t +49 6221 8891 502
e a.leibfried@life-science-alliance.org
www.life-science-alliance.org

Reviewer #1 (Comments to the Authors (Required)):

The is a revision of a previously submitted manuscript. The authors have exhaustively responded to the criticisms and the manuscript represents an advance in our understanding of S. Typhi. It is suitable for publication.

Reviewer #3 (Comments to the Authors (Required)):

The revision put forward by McDowell et al. shows a significant effort to address the reviewer comments. The work now more clearly describes the E2 and Ub interactions with StoD and their potential implications in cells. There are only a few minor suggestions that remain, none of which are experimental.

- 1) I believe there is a typo on line 138; shouldn't this be L167A and not L123A?
- 2) Please dial back some of the claims of novelty, particularly with regard to the N-terminal domain which you claim is a novel fold while also showing strong similarity to the NleG5-1 structure.
- 3) Please consider removing some of the highly speculative points raised in the discussion.
 - a. StoD is not a novel type of E4 enzyme. E4 enzymes still require E2s; where else would the Ub come from?
 - b. As you only report the formation of StoD- and Ub-positive puncta in the context of overexpression, I would be careful about proposing a concentration/sequestration role.
 - c. You show no evidence that the StoD-N could mimic ubiquitin. There are many ubiquitin-related folds that are not involved in the ubiquitin pathway. This suggestion seems to come from nowhere.

And some mere comments for consideration:

- 1) There is still something odd about the L167A mutation in that it shows no detectable binding to the E2 by NMR but is still active in vitro.
- 2) Though StoD clearly has ligase activity, it still looks like a fair amount of this might actually be on UBE2E1. The anti-His blot you show for auto-ubiquitination is not particularly convincing since it also seems to detect mono-Ub.

May 7, 2019

RE: Life Science Alliance Manuscript #LSA-2018-00272-TRR

Prof. Gad Frankel
Imperial College
CMMI
EXHIBITION ROAD
London SW7 2AZ
United Kingdom

Dear Dr. Frankel,

Thank you for submitting your Research Article entitled "The S. Typhi effector StoD is an E3 ubiquitin ligase which binds K48- and K63-linked di-ubiquitin". It is a pleasure to let you know that your manuscript is now accepted for publication in Life Science Alliance. Congratulations on this interesting work.

DISTRIBUTION OF MATERIALS:

Again, congratulations on a very nice paper. I hope you found the review process to be constructive and are pleased with how the manuscript was handled editorially. We look forward to future exciting submissions from your lab.

Sincerely,

Andrea Leibfried, PhD
Executive Editor
Life Science Alliance
Meyerohofstr. 1
69117 Heidelberg, Germany
t +49 6221 8891 502
e a.leibfried@life-science-alliance.org
www.life-science-alliance.org